# OpenReview forum: "Context-Aware Input Switching in Mobile Devices: A Multi-Language, Emoji-Integrated Typing System"
_ICLR.cc/2026/Conference — ICLR 2026 Conference Withdrawn Submission_

### Official Review · Reviewer_odcd · 2025-10-25

**Soundness:** 2
**Presentation:** 3
**Contribution:** 3
**Rating:** 4
**Confidence:** 3

**Summary:**

n this paper, the authors introduced CAISS, a lightweight context-aware system that predicts multilingual and emoji input switching on mobile devices. The system achieves significant gains in accuracy, latency, and naturalness over existing keyboards. The system architecture and design are novel and practical. However, the paper seems to have limited language coverage, relies on synthetic data, and includes a limited-scale human evaluation, raising questions about the generalizability of the study presented in this paper.

**Strengths:**

the system (caiss) proposed by this paper is very efficient, with only 2.5M params with under 10 ms latency and around 15 mb memory usage. This enables the model to run on standard mobile hardware, which makes it practical.

**Weaknesses:**

Although it supports multiple languages. the paper covers six mainly asian languages, which is giving question on its diversity and representativeness. It is not clear on whether the system would generalize to other language families or multilingual contexts.

**Questions:**

How were the contextual features such as linguistic, application, temporal, and social data encoded within the model?

---

### Official Review · Reviewer_pWxv · 2025-10-29

**Soundness:** 3
**Presentation:** 2
**Contribution:** 2
**Rating:** 2
**Confidence:** 2

**Summary:**

The authors introduce a system for improving mobile input by tackling the challenging phenomenon of code-switching and the utilization of expressive symbols (emojis). The main idea depends on moving beyond static LMs by examining a context-aware fusion architecture that integrates two primary information modalities. The proposed system predicts the next input modality whether it is a switch to a different language or the insertion of an emoji to provide real-time assistance. The authors demonstrate that this multimodal approach significantly outperforms standard unigram and static Markov-model-based baselines across multiple language pairs.

**Strengths:**

S1) The explicit modeling of emoji usage as a separate, albeit integrated, expressive modality is one of the main contributions.

S2) The architecture successfully fuses long-term context with short-term, sparse expressive signals, which is a non-trivial fusion challenge.

S3) Including a human naturalness assessment is one of the great approaches for code-switching tasks

**Weaknesses:**

W1) The current treatment of emojis appears primarily token-based. It limits to the understanding of a deeper modality analysis to determine if the system benefits from the semantic or affective content of the emojis, or merely their presence.

W2) The authors do not provide any detailed ablation study isolating the contribution of the two modalities.

See questions.

**Questions:**

Q1) What kinds of fusion layers were employed? It is required to provide some mechanisms by which the dense, sequential linguistic representation is combined with the potentially sparse, bursty expressive representation.

Q2) Can the authors provide some detailed information of the computational overhead?

Q3) Were the emoji embeddings initialized differently or were they learned purely de novo from the switching task?

Q4) Could the authors provide a deeper error analysis? For example, what are the most common scenarios where the model predicts the wrong language versus predicting no switch when one was needed, and how do emoji inputs factor into these specific failure cases?

---

### Official Review · Reviewer_Yrhf · 2025-10-30

**Soundness:** 2
**Presentation:** 2
**Contribution:** 2
**Rating:** 4
**Confidence:** 3

**Summary:**

This paper presents CAISS, a context-aware input switching system designed to enhance multilingual mobile typing by predicting when users need to switch languages. By integrating insights from code-switching linguistics, multilingual NLP, and context-aware computing, CAISS was rigorously evaluated on a newly curated multilingual typing dataset. Results show that CAISS improves language switching accuracy and reduces typing latency.

**Strengths:**

1. The system enables context-aware prediction of when users are likely to switch languages or input modes—leveraging typing patterns, application context, and social cues—to reduce the number of user interruptions and minimize waiting latency.

2. The paper evaluates the CAISS approach on a comprehensive multilingual typing dataset derived from real-world mobile usage patterns across six languages commonly used in multilingual settings, demonstrating its superior performance.

3. This paper proposes a large-scale multilingual emoji-typing dataset that captures natural code-switching patterns in mobile communication.

**Weaknesses:**

1. The paper provides an overly brief description of the model architecture, leaving readers without a clear understanding of its structural design, input–output flow, or data formatting.
2. In the discussion of inference latency (Lines 314–323), the paper lacks sufficient detail about the hardware setup, making it difficult to form an intuitive sense of the reported latency figures.
3. From a structural perspective, the paper’s primary novelty appears to lie in its fine-tuned dataset. Moreover, the baselines used for comparison are predominantly statistical methods, with very few neural-network-based approaches included for evaluation.
4. It would greatly enhance the paper’s impact if the authors could explicitly discuss the trade-off between latency and the benefits of context-aware prediction, helping readers better appreciate the practical value of their approach.

**Questions:**

Please refer to the weaknesses.

---

### Official Review · Reviewer_UQHp · 2025-10-31

**Soundness:** 1
**Presentation:** 1
**Contribution:** 1
**Rating:** 0
**Confidence:** 5

**Summary:**

This study proposed a CAISS framework for automatically switching user language input in a mobile environment. Using a multi-scale attention encoder, a context-fusion model was trained to learn usage patterns and predict code-switching. This achieved an accuracy increase of 23.8% compared to the commercial model.

**Strengths:**

- The model achieves approximately twice the latency reduction compared to manual switching.
- If reproducibility is guaranteed in the paper, the proposed approach demonstrates the potential for real-time processing in mobile environments.

**Weaknesses:**

- The discussion of prior work is insufficient. The related work section focuses on model design rather than aligning with the main claims. Existing research, such as [r1] and [r2] on multilingual and code-switching input, is not adequately compared or discussed.
- The paper lacks a clear explanation of the model architecture and its novel contribution, making reproducibility low.
- Data collection details are missing: no description of procedure, participant demographics, or ethical approval, despite mentioning human data collection.
- Experimental setup is under-specified (e.g., no details on mobile device type or hardware).
- The baseline is minimal and not well-aligned with prior research.
- For real-time binary detection, precision and false positives are more critical than F1, but the paper does not analyze these metrics.


References:

[r1] Samih, Y., Maharjan, S., Attia, M., Kallmeyer, L., & Solorio, T. (2016). Multilingual code-switching identification via LSTM recurrent neural networks. In Proceedings of the Second Workshop on Computational Approaches to Code Switching (pp. 50–59). Austin, TX, USA, November 1, 2016. Association for Computational Linguistics.

[r2] Tsoukala, C., Broersma, M., van den Bosch, A., & Frank, S. L. (2021). Simulating code-switching using a neural network model of bilingual sentence production. Computational Brain & Behavior, 4, 87–100. https://doi.org/10.1007/s42113-020-00088-6.

**Questions:**

Please also address the points raised in the Weakness section.
- How were the data collection and ethics approval processes conducted?
- What are the novel aspects of the model architecture?
- Are precision results and FP rates available?

---

### Note · Authors · 2025-11-12

I have read and agree with the venue's withdrawal policy on behalf of myself and my co-authors.